# Adaptive Test-Time Training for Predicting Need for Invasive Mechanical Ventilation in Multi-Center Cohorts

**Xiaolei Lu**
Divisions of Biomedical Informatics
University of California San Diego
La Jolla, CA 92122, USA
xil270@ucsd.edu

**Shamim Nemati** *
Divisions of Biomedical Informatics
University of California San Diego
La Jolla, CA 92122, USA
snemati@ucsd.edu

## Abstract

Accurate prediction of the need for invasive mechanical ventilation (IMV) in intensive care units (ICUs) patients is crucial for timely interventions and resource allocation. However, variability in patient populations, clinical practices, and electronic health record (EHR) systems across institutions introduces domain shifts that degrade the generalization performance of predictive models during deployment. Test-Time Training (TTT) has emerged as a promising approach to mitigate such shifts by adapting models dynamically during inference without requiring labeled target-domain data. In this work, we introduce Adaptive Test-Time Training (AdaTTT), an enhanced TTT framework tailored for EHR-based IMV prediction in ICU settings. We begin by deriving information-theoretic bounds on the test-time prediction error and demonstrate that it is constrained by the uncertainty between the main and auxiliary tasks. To enhance their alignment, we introduce a self-supervised learning framework with pretext tasks: reconstruction and masked feature modeling optimized through a dynamic masking strategy that emphasizes features critical to the main task. Additionally, to improve robustness against domain shifts, we incorporate prototype learning and employ Partial Optimal Transport (POT) for flexible, partial feature alignment while maintaining clinically meaningful patient representations. Experiments across multi-center ICU cohorts demonstrate competitive classification performance on different test-time adaptation benchmarks.

## 1 Introduction

Invasive mechanical ventilation (IMV) is a critical intervention utilized in intensive care units (ICUs) for patients with severe respiratory failure and acute respiratory distress syndrome (ARDS) (Grotberg et al., 2023). However, its use is complicated by the risk of ventilator-induced lung injury and complications resulting from prolonged IMV. Timely and accurate identification of patients at high risk for mechanical ventilation is crucial for optimizing clinical decision-making. Early recognition of these patients enables proactive medical interventions and facilitates efficient resource allocation within hospital system (Fan et al., 2018).

In recent years, the development of machine learning (ML) models has shown great promise in predicting the need for IMV, which leverages electronic health record (EHR) data to identify complex patterns that human clinicians might overlook (Shashikumar et al., 2021b). These models can incorporate diverse features, including vital signs and laboratory results to enhance prediction accuracy and provide critical decision support in ICU settings. However, the effective deployment of such models in real-world clinical settings remains a challenge. A key issue is the variability in data distributions across hospitals due to differences in patient populations, clinical practices, and EHR systems. These shifts, often referred to as domain shifts, can substantially degrade the performance of predictive models that were trained on data from a single or limited number of sources. For in-

---

*Corresponding author. Email: snemati@ucsd.edu

stance, a respiratory failure prediction model (Lam et al., 2024) trained on ICU cohort from UC San Diego Health showed an approximately 12% drop in the area under the curve (AUC) when evaluated on an external ICU cohort.

Addressing this challenge requires adaptive methodologies that can account for site-specific heterogeneity. Existing approaches include pre-training models on large multi-center datasets (Shashikumar et al., 2021b), and transfer learning to fine-tune models on site-specific data (Lam et al., 2024) to align feature representations across domains. While these methods have shown promise, many require access to labeled data from the target domain during training or involve computationally expensive retraining processes, which are not always feasible in real-time clinical settings. Test-time training (TTT) offers a novel and efficient solution to this problem by enabling models to adapt dynamically at the time of prediction, without requiring pre-access to target domain labels or costly re-training. TTT leverages an auxiliary task, trained alongside the main task, to update the model's parameters or representations using the test input itself.

Predictive systems based on EHR data, such as Composer (Shashikumar et al., 2021a), have shown significant real-world impact to improve clinical outcomes through real-time decision support (Boussina et al., 2024). In a before-and-after quasi-experimental design study at two emergency departments (EDs), the Composer model for sepsis prediction significantly increased bundle compliance and reduced in-hospital mortality. However, limited prior work has explored TTT in the context of EHR data, particularly in real-world clinical scenarios. By enabling dynamic adaptation at prediction time, TTT addresses variability across institutions and patient populations, ensuring robust performance in critical tasks, such as predicting IMV need in multi-center cohorts. Unlike pretraining/offline alignment approaches (e.g., MaskTab (Chen et al.), PhyMask (Kara et al., 2024), and SPOT (Gurumoorthy et al., 2021)), which require source data and extended training, our setting is source-free and demands on-the-fly adaptation at deployment.

In this study, we introduce Adaptive Test-Time Training (AdaTTT) for predicting IMV need 24 hours in advance in ICU patients across multi-center cohorts. AdaTTT is designed to address domain shifts in EHR data through adaptive self-supervised learning and robust feature alignment. Our key contributions are as follows:

- We derive information-theoretic bounds on the test-time prediction error to show that the error is constrained by the uncertainty between the main and auxiliary tasks, which guide the design of auxiliary tasks for better adaptation.

- We introduce two SSL tasks: Reconstruction and Masked Feature Modeling along with a dynamic masking strategy that prioritizes the most informative features during test-time training. The masking probabilities adapt based on feature relevance to the primary task, ensuring that the SSL task remains aligned with the IMV prediction objective.

- To prevent overfitting to individual test samples, we integrate prototype learning with Partial Optimal Transport (POT) to allow partial matching between source domain features and test-time distributions, which promotes robust generalization while avoiding excessive adaptation to test-domain noise.

- We conduct extensive experiments on multi-site ICU cohorts, where our method achieves competitive classification performance across various test-time adaptation benchmarks.

## 2  RELATED WORK

### 2.1  PREDICTIVE MODELS FOR INVASIVE MECHANICAL VENTILATION

Early IMV-risk tools such as ROX and regression scores are interpretable but struggle with nonlinear, time-varying physiology (Roca et al., 2019). Leveraging EHR-scale data, VentNet predicts IMV 24h ahead with a feedforward model (Shashikumar et al., 2021b); encoder–decoder designs like DBNet integrate structured signals and demographics (Zhang et al., 2021); and multimodal hybrids that fuse CXR with EHR further boost discrimination (Tandon et al., 2023). However, cross-site performance often degrades due to population, workflow, and EHR heterogeneity; recovery via target-domain fine-tuning is common but label-intensive and operationally impractical for continuous deployment.

## 2.2 TEST-TIME ADAPTATION

TTA adapts models on unlabeled test inputs without revisiting source data (Liang et al., 2024). Batch normalization(BN)-centric methods include prediction-time BN statistics updates (Nado et al., 2020) and TENT's entropy minimization for BN parameters (Wang et al., 2020), while source-free SHOT freezes the classifier and adapts the encoder with pseudo-labels (Liang et al., 2020). Test-time training (TTT) attaches auxiliary SSL branches for online encoder updates (Sun et al., 2020); extensions like TTT++ (contrastive) and ClusT3 (clustering) improve alignment but may inherit instability or assume domain consistency (Liu et al., 2021; Hakim et al., 2023).

Three relevant directions are: (i) T3A, an optimization-free method forming class prototypes from streaming test data to reweight logits that is efficient but classifier-level only, assuming stable class structure (Iwasawa & Matsuo, 2021); (ii) SAR, which filters unreliable samples and applies sharpness-aware entropy minimization for stable BN updates that is effective with small batches yet BN-dependent (Niu et al., 2023); (iii) CoTTA, maintaining a moving teacher with augmentation- and weight-averaged pseudo-labels plus periodic restoration, is useful for long horizons but hyperparameter-sensitive with potential error accumulation (Wang et al., 2022).

In EHR-driven IMV prediction, these approaches face practical challenges: tiny per-encounter batches undermine BN estimates; pseudo-labeling struggles with class imbalance and temporal non-stationarity; clustering assumptions break under irregular sampling and missingness; classifier-only adaptation cannot address representation shift, making direct application from vision to ICU EHRs difficult.

# 3 METHODOLOGY

## 3.1 PRELIMINARY: TEST-TIME TRAINING

Let $x \in \mathcal{X}$ denote an input instance from the covariate space, $y_m \in \mathcal{Y}_m$ denote the corresponding label for the main task (e.g., classification), and $y_s \in \mathcal{Y}_s$ denote the auxiliary label derived for a self-supervised task. The training set is represented as $\{(x_i, y_{m,i})\}_{i=1}^{n_s}$. At test time, both covariate distribution $p(X')$ and label distribution $p(Y'_m) \, p(Y'_s)$ may change, which leads to domain shifts.

Test-Time Training (Sun et al., 2020) addresses these shifts by leveraging the same SSL task during both the training and testing phases to align features between the training domain and individual test instances. The framework consists of a shared feature encoder $f_e(\cdot; \theta_e)$, a primary classification head $h_c(\cdot; \theta_c)$ and an SSL head $h_s(\cdot; \theta_s)$.

During training, TTT jointly optimizes both the main classification loss $\mathcal{L}_{\mathrm{main}}$ and the auxiliary SSL loss $\mathcal{L}_{\mathrm{ssl}}$ as

$$\theta_e^*, \theta_c^*, \theta_s^* = \arg \min_{\theta_e, \theta_c, \theta_s} \sum_{i=1}^{n_s} \mathcal{L}_{\mathrm{main}}(x_i, y_i; \theta_e, \theta_c) + \mathcal{L}_{\mathrm{ssl}}(x_i; \theta_e, \theta_s). \tag{1}$$

At inference time, rather than relying on static model parameters, TTT dynamically adapts the encoder for each test instance $x'$ by optimizing the SSL objective with

$$\theta_e(x') = \arg \min_{\theta_e} \mathcal{L}_{\mathrm{ssl}}(x'; \theta_s^*, \theta_e). \tag{2}$$

The adapted encoder is then used to obtain the final prediction with

$$\hat{y} = h_c(f_e(x'; \theta_e(x')); \theta_c^*). \tag{3}$$

## 3.2 THEORETICAL INSIGHTS

Prior work (Liu et al., 2021) derives accuracy bounds under assumptions of distributional alignment and task consistency. We provide an independent perspective based on information theory that examines how the auxiliary task informs the main task through shared representations. Let $Z$ and $Z'$ represent the feature representations from the feature encoder for the training and test domains,

respectively. We define $\pi_m$ is the main task classifier predicting $Y_m$, and $\pi_s$ is the SSL classifier predicting $Y_s$. The probability $P(\pi_m(Z') = Y'_m)$ quantifies the likelihood that the main task classifier $\pi_m$ correctly predicts the main task label $Y'_m$ at test time.

In test-time training, we assume the Markov chain $Y'_s \to Z' \to Y'_m$ holds, which captures the dependency structure where the auxiliary task labels $Y'_s$ influence the main task labels $Y'_m$ only through the shared representation $Z'$. Under this assumption, we establish the relationship between the mutual information of the auxiliary and main tasks (please refer to Appendix A.1 for derivation details).

In test-time training, where only the shared representation layers are updated using $Y'_s$, the following inequality holds:

$$I(Z'; Y'_m) \geq I(Y'_s; Y'_m). \tag{4}$$

Building on this, we derive information-theoretic bounds on the main task prediction error with binary case in the ideal scenario (please refer to Appendix A.2 for derivation details and multi-class case). Let $\eta(z') = P\{Y'_m = 1 \mid Z' = z'\} > 0.5$ as positive, the minimum classification error is $p(e) = \int_{Z'} \min\{\eta(z'), 1 - \eta(z')\}dp(z')$ and $H_{\text{err}}(\eta) = -\eta \cdot \log \eta - (1 - \eta) \cdot \log(1 - \eta)$, the prediction error is bounded by

$$H_{\text{err}}^{-1}(H(Y'_m \mid Y'_s)) \leq p(e) \leq \frac{1}{2} H(Y'_m \mid Y'_s). \tag{5}$$

The lower and upper bounds on the prediction error of the main task highlights the relationship between the main task and the SSL task in performance after adaptation. The upper bound shows that error is limited by the conditional uncertainty $H(Y'_m \mid Y'_s)$ while the lower bound demonstrates that lower $H(Y'_m \mid Y'_s)$ improves worst-case guarantees. Additionally, under domain shift $w(y_s) = \frac{P(Y'_s = y_s)}{P(Y_s = y_s)}$, $H(Y'_m \mid Y'_s) = \sum_{y_s} w(y_s)P(Y_s = y_s)H(Y'_m \mid Y'_s = y_s)$, Theorem 1 shows overfitting to the test-domain auxiliary task distribution $P(Y'_s)$ can lead to overweighting regions with high uncertainty $H(Y'_m \mid Y'_s)$. Enforcing $P(Y'_s) = P(Y_s)$ without accounting for test-specific shifts can further harm model generalization. These findings emphasize the necessity of designing a framework that ensures strong alignment between the main and auxiliary tasks while remaining robust to domain shifts for effective test-time adaptation.

## 3.3 Adaptive Test-Time Training

The effectiveness of test-time training depends on SSL alignment with the main task and handling distribution shifts. To address this, we propose Adaptive Test-Time Training (AdaTTT) to enhance TTT with dynamic self-supervised learning and prototype-guided adaptation, which aims to improve generalization under clinical domain shifts in EHR data.

### 3.3.1 Dynamic self-supervised learning

Fixed SSL transformations (e.g., random feature masking) may introduce spurious patterns unrelated to the main task. In EHR data, some features are more predictive, and treating all equally reduces adaptation effectiveness. To mitigate this, we introduce two pretext tasks: Reconstruction and Masked Feature Modeling along with a dynamic feature masking strategy that prioritizes informative features.

**SSL Loss**. Given an input vector x and the the corrupted input x̃, the reconstruction loss and masked feature modeling loss are defined as

$$\mathcal{L}_{\text{recon}} = \frac{1}{d} \sum_{j=1}^{d} (x_j - \hat{x}_j)^2, \tag{6}$$

where $d$ is the total number of features, and $\hat{x}_j$ represents the reconstructed value of feature $x_j$ predicted by the model.

$$\mathcal{L}_{\text{mfm}} = \frac{1}{|M|} \sum_{j \in M} (x_j - \hat{x}_j)^2, \tag{7}$$

where $M$ is the set of indices of masked features ($m_j = 1$), and $|M|$ denotes the number of masked features.

The overall self-supervised learning loss combines the reconstruction loss and the masked feature modeling loss with

$$\mathcal{L}_{\text{ssl}} = \lambda_{\text{recon}} \cdot \mathcal{L}_{\text{recon}} + \mathcal{L}_{\text{mfm}}, \tag{8}$$

**Dynamic Feature Masking**. Instead of fixed random masking, we introduce an adaptive masking strategy that assigns higher masking probabilities to more informative features. Given an input vector $\mathbf{x}$, the corrupted input $\tilde{\mathbf{x}}$ is generated as follows:

$$\tilde{x}_j = m_j \cdot f(\mathbf{x}, j) + (1 - m_j) \cdot x_j, \tag{9}$$

where $\mathbf{m} \in [0,1]^d$ is the mask vector with elements $m_j$. $f(\mathbf{x}, j) \sim P(x_j)$ is a replacement for feature $x_j$ where $P(x_j)$ is the empirical distribution of training or testing data.

Masking probabilities are dynamically updated based on global feature relevance scores derived from the main task:

$$I_j = \frac{1}{n_s} \sum_{n=1}^{n_s} \left| \frac{\partial Y_m^{(n)}}{\partial x_j^{(n)}} \cdot x_j^{(n)} \right|, \tag{10}$$

$$p_{m,j} = \frac{I_j - \min_k I_k}{\max_k I_k - \min_k I_k}. \tag{11}$$

The masking phrases in the training section are described as follows: In the training phase, we employ a two-stage masking strategy. During the warmup phase (Epochs 1–N), dynamic masking is applied using a fixed prior mask probability to encourage broad feature exploration (the prior is derived from a pretrained respiratory failure prediction model). In the subsequent adaptive masking phase (Epochs N+1–End) feature relevance scores are updated at each epoch based on the model's main task predictions from the previous epoch. These relevance scores are then used to refine the masking probabilities to enable the model to focus on more informative features over time.

During test-time training, the model continues refining masking probabilities at each gradient step to ensure SSL tasks remain aligned with the primary task. Prioritizing the most informative features challenges the model to reconstruct or predict essential aspects of the data and then improve generalization under domain shifts.

### 3.3.2 PROTOTYPE-GUIDED ADAPTATION

Prototype-guided adaptation is particularly important in the context of EHR-based test-time training. Instance-level SSL updates may overfit to noisy or highly site-specific measurements, especially when the target-domain distribution differs substantially from the source. To mitigate this, we introduce a set of learned prototypes that capture population-level structure from the source domain. These prototypes act as stable anchors during adaptation. By leveraging Partial Optimal Transport (POT), the model can perform soft and partial alignment between test-time features and prototype distributions, enabling flexible yet robust adaptation under domain shift while avoiding the brittleness of full-transport matching.

**Training Stage**. We introduce a prototype learning loss that encourages the shared layer features $z$ to align with their corresponding prototypes $\mathbf{P}$. These prototypes are designed to effectively represent the distribution of the feature space $z$ and ensure that the feature embeddings are compact and structured around these representative points.

The prototype learning loss is defined as

$$\mathcal{L}_{\text{proto}}(z_i; \mathbf{P}) = \|z_i - p_{\mathcal{A}(z_i)}\|_2^2, \tag{12}$$

where $\mathbf{P} = \{p_1, p_2, ..., p_k\}$ is the set of $k$ prototypes. $p_{\mathcal{A}(z_i)}$ is the prototype assigned to the feature $z_i$ based on the cluster assignment $\mathcal{A}(z_i)$.

To prevent all features are assigned to a single prototype, a regularization term is added to balance the cluster assignments:

$$\mathcal{L}_{\text{reg}}(\mathbf{P}) = \sum_{j=1}^{k} \left( \frac{1}{n_s} \sum_{i=1}^{n_s} \mathbb{I}(\mathcal{A}(z_i) = j) - \frac{1}{k} \right)^2, \tag{13}$$

where $\mathbb{I}(\mathcal{A}(z_i) = j)$ is an indicator function that equals 1 if $z_i$ is assigned to prototype $p_j$.

The final loss function incorporating prototypes is given as

$$\mathcal{L} = \sum_{i=1}^{n_s} \Big[ \mathcal{L}_{\text{main}}(x_i, y_i) + \mathcal{L}_{\text{ssl}}(x_i) + \lambda_{\text{proto}} \mathcal{L}_{\text{proto}}(z_i) \Big] + \lambda_{\text{reg}} \mathcal{L}_{\text{reg}}(\mathbf{P}), \tag{14}$$

where $\mathcal{L}_{\text{main}}$ is the respiratory failure prediction loss. $\mathcal{L}_{\text{ssl}}$ is the self-supervised loss (see Section 3.3.1). $\mathcal{L}_{\text{reg}}$ is the regularization loss for balanced assignment. $\lambda_{\text{proto}}$ and $\lambda_{\text{reg}}$ are hyperparameters controlling the importance of the prototype and regularization terms.

**Test-Time Training Stage**. Traditional Optimal Transport (OT) assumes a full alignment between source and target distributions, which may be too rigid in the presence of domain shifts. We refine the alignment between the training prototypes $\mathbf{P}$ and the test-time feature representations $z'$ by incorporating POT. Instead of constraining the transport plan to only partially align prototypes with the test instance (Chapel et al., 2020), we augment the set of $\mathbf{z}'$ by adding $k-1$ perturbed duplicates to transform the transport problem into a standard optimal transport setting while enabling partial matches between $z'$ and the prototypes $\mathbf{P}$.

$$\mathbf{z}' = \{z', z_1', \dots, z_{k-1}'\}, \tag{15}$$

where each duplicate $z_j'$ is sampled as

$$z_{j,d}' \sim z_d' + \mathcal{N}(0, \sigma_d^2), \tag{16}$$

$$\sigma_d^2 = \frac{1}{k} \sum_{j=1}^{k} (p_{j,d} - \mu_d)^2, \tag{17}$$

where $p_{j,d}$ is the value of the $d$-th dimension of the $j$-th prototype $\mathbf{p}_j$, $\mu_d$ is the mean of the $d$-th dimension across all prototypes.

The loss function employed during test-time training is defined as

$$\mathcal{L}_{\text{test}} = \mathcal{L}_{\text{ssl}} + \lambda_{\text{ot}} \cdot \sum_{i,j} \gamma_{ij} C_{ij}, \tag{18}$$

where $\lambda_{\text{ot}}$ is a hyperparameter balancing the importance of the OT cost in the overall loss. $\gamma_{ij}$ defines the mass transported from $\mathbf{z}_i'$ to the $j$-th prototype. $C_{ij} = \|\mathbf{z}_i' - \mathbf{p}_j\|_2^2$ represents the squared Euclidean distance between $\mathbf{z}_i'$ and the prototype $\mathbf{p}_j$.

## 4 Experiments

### 4.1 Experimental Setting

**Datasets**. We conduct a retrospective study using de-identified EHR data of all adult patients ($\geq 18$ years) admitted to the ICU at UCSD Health[1] between January 1, 2016, and December 31, 2023. This dataset served as the development and validation cohort. To evaluate the mechanisms of test-time training, we utilize additional datasets, including ICU admissions at UCSD Health between January 1, 2024, and June 30, 2024, UCI Health between January 1, 2023, and August 31, 2024, as well as the publicly available MIMIC-IV dataset. Institutional Review Board approval was obtained for the use of these datasets under protocol #800258 ("A Real-Time Multimodal Data Integration Model for Prediction of Respiratory Failure in Patients with COVID-19"). Appendix B.1 provides full details on cohort selection and data processing. Our development cohort consists of 24,943 encounters, with 1,308 positive cases (IMV prevalence: 5.2%). The testing cohorts include UCSDH (1,835 encounters, 104 positive cases, IMV prevalence: 5.7%) and UCIH (2,564 encounters, 141 positive cases, IMV prevalence: 5.5%). The original MIMIC-IV dataset contains 35,534 encounters with an IMV prevalence of 15.4%. For computational efficiency, we randomly downsampled MIMIC-IV to 2,069 encounters (244 positive cases with an IMV prevalence of 11.8%).

**Implementation Details**. Our network architecture follows Vent.io (Lam et al., 2024) (Appendix B.2). We use Bayesian optimization for source-domain pretraining to tune general network

---

[1]For brevity, we will hereafter refer to UCSD Health and UCI Health as UCSDH and UCIH, respectively.

Table 1: AUC (%) across testing sites (↑ higher is better).

| Dataset | TEST | CoTTA | T3A | SAR | TENT | TTT | TTT++ | ClusT3 | NC-TTT | PriTTT (Ours) | DynTTT (Ours) | AdaTTT (Ours) |
|---|---|---|---|---|---|---|---|---|---|---|---|---|
| UCSDH | 84.01 | 83.12±0.02 | 82.50±0.04 | 84.30±0.04 | 82.19±0.11 | 82.55±0.09 | 82.50±0.06 | 82.36±0.08 | 82.32±0.06 | 84.61±0.03 | 84.54±0.10 | **85.02±0.05** |
| UCIH | 83.75 | 83.81±0.04 | 83.10±0.05 | 83.20±0.10 | 83.06±0.08 | 82.81±0.05 | 82.85±0.10 | 81.99±0.12 | 83.62±0.10 | 83.98±0.06 | 83.84±0.12 | **84.10±0.05** |
| MIMIC-IV | 75.28 | 76.60±0.05 | 76.10±0.03 | 75.72±0.04 | 74.34±0.05 | 76.45±0.07 | 76.24±0.08 | 74.41±0.11 | 75.27±0.08 | 76.84±0.03 | 76.79±0.05 | **77.17±0.08** |

Table 2: Brier score across testing sites (↓ lower is better).

| Dataset | TEST | CoTTA | T3A | SAR | TENT | TTT | TTT++ | ClusT3 | NC-TTT | PriTTT (Ours) | DynTTT (Ours) | AdaTTT (Ours) |
|---|---|---|---|---|---|---|---|---|---|---|---|---|
| UCSDH | 0.089 | 0.092±0.01 | 0.093±0.01 | 0.089±0.02 | 0.090±0.04 | 0.093±0.03 | 0.094±0.01 | 0.090±0.03 | 0.090±0.01 | **0.089±0.02** | 0.089±0.01 | **0.086±0.02** |
| UCIH | 0.089 | 0.091±0.01 | 0.091±0.02 | 0.091±0.02 | 0.091±0.03 | 0.094±0.02 | 0.095±0.04 | 0.092±0.04 | 0.091±0.03 | **0.090±0.03** | 0.089±0.02 | **0.085±0.02** |
| MIMIC-IV | 0.111 | 0.110±0.02 | 0.110±0.03 | 0.111±0.03 | 0.114±0.04 | 0.110±0.04 | 0.111±0.05 | 0.113±0.04 | 0.113±0.05 | **0.110±0.05** | 0.112±0.04 | **0.106±0.04** |

hyperparameters (learning rate, weight regularization, number of hidden layers). The prototype set size is $k = 4$. For dynamic masking, we adopt a two-phase schedule: a warm-up phase with a *fixed prior mask probability* of 0.5, followed by an adaptive phase where masking probabilities are updated from feature relevance. During deployment, test-time training (TTT) performs five gradient steps per input (we refer to each step as an "**iteration**"), and we follow the standard reset protocol (Sun et al., 2020) by restoring encoder weights to the pretrained state after each instance. For partial optimal transport, we use Sinkhorn with entropic regularization $\varepsilon = 0.1$, a maximum of 1000 Sinkhorn iterations, and a mini-batch size equal to the prototype count ($K = k$). All baselines are re-implemented with the same encoder, data pipeline, and hyperparameter search budget to ensure a fair comparison. [2]

**Evaluation**. We follow the clinical labeling scheme (Lam et al., 2024) to capture the physiological states of respiratory failure, defining score $\geq 3$ as positive and $< 3$ as control (Appendix B.1). Encounters are categorized as True Positive, False Positive, True Negative, or False Negative based on predictions within the specified prediction window (criteria in Appendix B.1). Performance is reported as AUC with mean $\pm$ standard error over 20 independent test-time training runs. In addition, following best practices for clinical prediction model assessment (Huang et al., 2020), we report Brier score for all models to evaluate calibration and clinical utility more comprehensively.

## 4.2 COMPARISON WITH BASELINES

**Baselines**. We consider a set of foundational and representative TTT and TTA methods and adapt each to the EHR domain to ensure fair and meaningful comparison in our experimental evaluations (Appendix B.3 provides full details of adapting baselines to EHR setting): **TEST**, **TENT** (Wang et al., 2020), **TTT** (Sun et al., 2020), **TTT++** (Liu et al., 2021), **ClusT3** (Hakim et al., 2023), **NC-TTT** (Osowiechi et al., 2024), **T3A** (Iwasawa & Matsuo, 2021), **SAR** (Niu et al., 2023) and **CoTTA** (Wang et al., 2022). We also evaluate two ablated versions of our method: **PriTTT**, which removes the adaptive distribution matching module and relies solely on updates to the mask probabilities for test-time training. **DynTTT**, which removes the dynamic masking module and focuses only on adaptive distribution matching.

**Results & Analysis.** Tables 1 (AUC) and 2 (Brier) report discrimination and calibration across three cohorts. AdaTTT obtains the top AUC on every site. Among non-TTT/TTA methods, SAR is closest on UCSDH (84.30±0.04), and CoTTA is competitive on MIMIC-IV (76.60±0.05), yet both remain below AdaTTT. Regarding calibration, AdaTTT improves or matches calibration on UCSDH/UCIH (Brier **0.086/0.085** vs. TEST 0.089/0.089) and achieves 0.106±0.04 on MIMIC-IV.

The behavior of the baselines is consistent with their assumptions and with the numbers in Tables 1–2. TENT's entropy minimization encourages confidence that can be misplaced on out-of-distribution patients. Standard TTT and TTT++ attach feature-agnostic SSL objectives and, in the

---

[2]Clean version implementation https://github.com/StillLu/AdaTTT.

latter, enforce relatively rigid source–target alignment; both are brittle when clinical feature importance is highly unequal across variables and when hospital shifts are complex. ClusT3's discrete codes and domain-consistency assumption struggle with continuous physiology and site heterogeneity; NC-TTT's contrastive likelihoods depend on well-specified "noise," which is hard to define for heterogeneous EHR features. Classifier-only adjustment in T3A helps when classes are tightly clustered (MIMIC-IV 76.10±0.03) but cannot correct representation shift (UCSDH 82.50±0.04). SAR stabilizes BN-based updates and fares well on UCSDH (84.30±0.04) but remains sensitive to batch-statistics quality, yielding inconsistent improvements on UCIH/MIMIC-IV.

We also examine how AdaTTT achieves its gains. Figure 1 illustrates the evolution of risk scores across time points for a patient from UCSDH. Early predictions trend lower, but risk escalates approaching the intubation event (within 24 hours), demonstrating dynamic response to clinical deterioration. First, aligning the auxiliary SSL objective with the clinical endpoint is critical: with dynamic, feature-aware masking, the SSL branch prioritizes clinically salient variables and downweights weak signals. Figure 2 traces

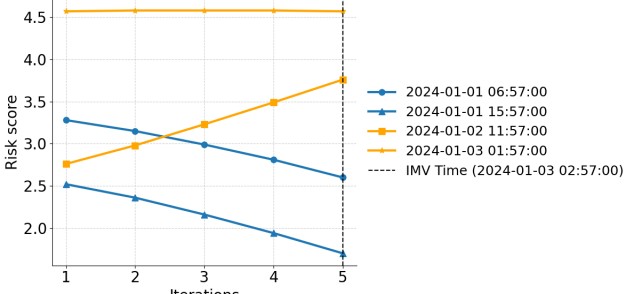

Figure 1: Risk score evolution during test-time training for a patient from UCSDH. Risk increases as intubation nears, which reflects model adaptation.

feature importance from pretrained priors through early to late training epochs[3]. During warm-up, masking follows the priors; as training proceeds, the distribution stabilizes and aligns with learned relevance, indicating tighter coupling between SSL and IMV prediction than in feature-agnostic TTT/TTT++. Second, at deployment the model refines importance *per input*: Figure 3 shows that some features remain stable while others (e.g., respiratory rate) gain weight across iterations, consistent with model faithfulness and clinical plausibility. In parallel, prototype-guided partial optimal transport flexibly matches test-time representations to learned prototypes, limiting overfitting to idiosyncratic or noisy samples while preserving clinically meaningful structure (see Appendix C.1 for an illustrative alignment). Ablations support this interpretation: PriTTT (feature-aware masking only) and DynTTT (distribution matching only) each improve over TTT/TTT++, but their combination yields the most consistent AUC and Brier gains across sites.

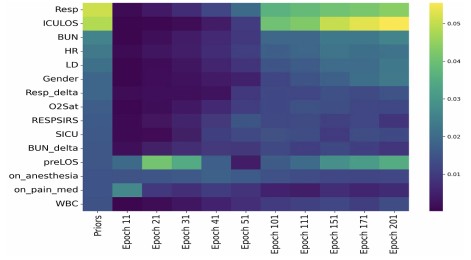

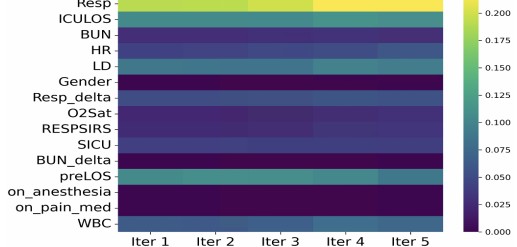

Figure 2: Feature importance evolution during training. The heatmap shows the changes in feature importance in the initial epochs and final epochs.

Figure 3: An example of feature importance evolution during test-time training. The heatmap shows the changes in feature importance across different iterations.

**Computational Cost**. Our proposed AdaTTT framework incorporates both feature importance updates and optimal transport computation during the test-time training phase. These additional operations inevitably increase the computational cost compared to standard TTT frameworks. We use the Sinkhorn algorithm (Sinkhorn, 1967) that leverages entropy regularization to ensure scalable computations. We evaluate the execution time of a single gradient update during test-time training. The average execution time is 0.29s, and did not change much, remaining at 0.26s when increasing the prototype size from 4 to 16.

---

[3]ICULOS: ICU length-of-stay to time $t$; Resp: respiratory rate; HR: heart rate; LD: lymphocyte differential; BUN: blood urea nitrogen; $O_2$Sat: oxygen saturation; WBC: white blood count.

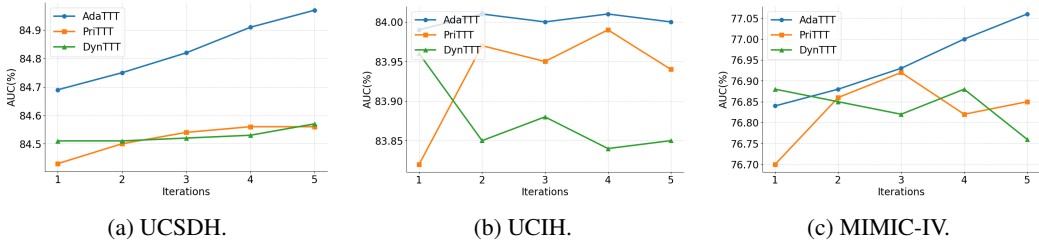

|     |     |     |
| --- | --- | --- |
| (a) UCSDH. | (b) UCIH. | (c) MIMIC-IV. |

Figure 4: Evaluation of the number of gradient updates for test-time training on different test cohorts.

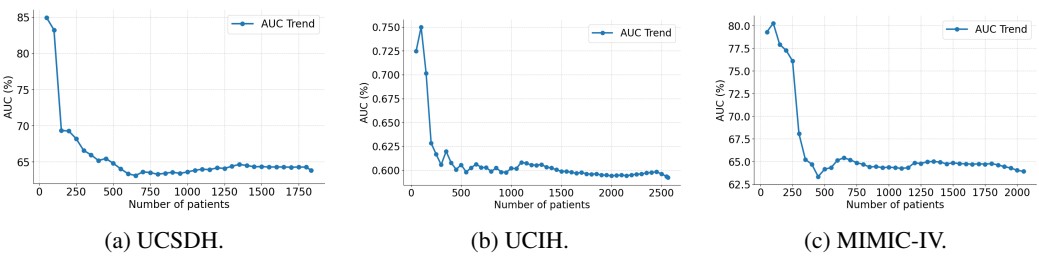

|     |     |     |
| --- | --- | --- |
| (a) UCSDH. | (b) UCIH. | (c) MIMIC-IV. |

Figure 5: Cumulative AUC trend over an increasing number of patients.

### 4.3 SENSITIVITY ANALYSIS

In this section, we examine the effect of the number of test-time training iterations and compare reset versus sequential update mechanisms. Additional ablations are provided in Appendix C.2 which investigate (1) the impact of prototype size, (2) the role of prototype learning, and (3) the effect of dynamic masking and the two SSL objectives.

**Comparison of the number of iterations**. Figure 4 presents the impact of the number of test-time training iterations on model performance across three different test cohorts. We evaluate the AUC scores as the number of gradient updates increases from 1 to 5 iterations. Across all sites, AdaTTT exhibits a stable and consistent improvement in AUC with more iterations. In contrast, PriTTT and DynTTT show less consistent trends, with fluctuations in performance, particularly in UCIH. PriTTT updates feature importance dynamically without a stable reference and is sensitive to initial feature importance variations. Meanwhile, DynTTT lacks feature selection control, which can lead to suboptimal emphasis on features between the main and SSL tasks.

**Reset versus Sequential Update Mechanism.** Compared with the reset strategy, the sequential update mechanism applies one gradient step at each new data point and retains the updated parameters across subsequent data points. Figure 5 presents the cumulative AUC trend across different sites under the sequential update mechanism. In UCIH and MIMIC-IV, performance initially improves as the model adapts to recent distributional shifts (e.g., achieving 80.27% AUC on MIMIC-IV), demonstrating the benefits of short-term adaptation. However, as updates continue across a larger patient population, the model's performance gradually deteriorates. This degradation is likely due to accumulated adaptation noise, which shifts the model's focus away from its originally learned feature structure.

## 5 CONCLUSION

In this study, we introduce AdaTTT, a lightweight and adaptive test-time training framework for predicting the need for invasive mechanical ventilation (IMV) 24 hours in advance across multi-center ICU cohorts. AdaTTT integrates feature-aware self-supervision with partial OT–based distribution alignment to address domain shift that naturally arises in real-world EHR time-series data. Across all external cohorts, the method yields consistent improvements in discrimination and calibration over strong TTT baselines, demonstrating practical value for real-time clinical risk monitoring.

To better contextualize these gains, we have added (1) an analysis of calibration and clinical relevance, and (2) additional experiments on secondary outcomes including acute kidney injury (AKI) and in-hospital mortality, showing similar trends.

We also explicitly acknowledge limitations: the method may be sensitive to extreme conditional shifts in $p(Y \mid X)$ (e.g., changes in clinical practice patterns), and prototype-based adaptation adds moderate computational overhead at test time. Addressing these challenges is an important direction for future work.

## ETHICS STATEMENT

**Compliance and oversight.** This study analyzes de-identified electronic health records (EHR) from multiple partner institutions (anonymized as Sites A/B) under appropriate institutional review and data-governance oversight, with a waiver of consent where applicable due to de-identification and minimal risk. All activities complied with relevant privacy regulations (e.g., HIPAA) and local security policies. No attempt was made to re-identify individuals, and all reported results are aggregate.

**Intended use and clinical safety.** The model is a research prototype for risk stratification and is *not* a stand-alone medical device. It should only be used with clinician oversight. Any real-world deployment would require prospective evaluation, safety monitoring, and regulatory review. We report discrimination and calibration (AUC, Brier score) to support assessment of clinical utility and risk.

**Fairness and shift.** We evaluate across distinct clinical sites to assess robustness under distribution shift, and we report calibration as recommended for clinical prediction models. Despite these efforts, residual bias and under-representation are possible; models trained in one setting may underperform elsewhere. We caution against out-of-scope use.

**Source-free adaptation safeguards.** Test-time adaptation updates only the encoder on a per-encounter basis and then resets weights before the next patient/time point, preventing cross-patient carryover. No patient-level exemplars or gradients are stored; adaptation logs contain no protected health information.

**Transparency and conflicts.** We will release code and configuration sufficient for reproduction (subject to data-use constraints). Funding and potential conflicts will be disclosed in the camera-ready version. All authors adhere to the ICLR Code of Ethics.

## REPRODUCIBILITY STATEMENT

We describe the cohort construction, preprocessing, and labeling scheme in Sec. B.1; the architecture and training protocol in Sec. B.2 and Sec. 4.1; and sensitivity analyses/ablations in Sec. C.2. We report full metric definitions (AUC, Brier) and evaluation procedures. We will release anonymized code (data loaders, feature engineering, training/evaluation scripts, and plotting utilities) to allow end-to-end replication with public data, and provide configuration files to reproduce all tables/figures from logs.

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

## A   THEORETIC ANALYSIS

### A.1   PROOF OF $I(Z'; Y'_m) \geq I(Y'_s; Y'_m)$

Using the chain rule of mutual information, we can expand the mutual information between $Z'$ and the joint variables $(Y'_m, Y'_s)$ as

$$I(Z'; Y'_m, Y'_s) = I(Z'; Y'_s) + I(Z'; Y'_m \mid Y'_s). \tag{19}$$

Using the chain rule of mutual information again, we have

$$
\begin{aligned}
I(Z'; Y'_m) &= I(Z'; Y'_m, Y'_s) - I(Z'; Y'_s \mid Y'_m) \\
&= \big(I(Z'; Y'_s) + I(Z'; Y'_m \mid Y'_s)\big) - I(Z'; Y'_s \mid Y'_m).
\end{aligned}
\tag{20}
$$

In the test-time training framework, $Z'$ is optimized to retain information from $Y'_s$ that is predictive of $Y'_m$. We model this by assuming

$$I(Z'; Y'_m \mid Y'_s) - I(Z'; Y'_s \mid Y'_m) \geq 0.$$

Applying the Data Processing Inequality (DPI) to our Markov chain, we obtain

$$I(Y'_s; Y'_m) \leq I(Y'_s; Z'). \tag{21}$$

From the mutual information decomposition derived earlier, we know that

$$I(Z'; Y'_m) \geq I(Z'; Y'_s) + I(Z'; Y'_m \mid Y'_s). \tag{22}$$

Since mutual information is always non-negative,

$$I(Z'; Y'_m \mid Y'_s) \geq 0. \tag{23}$$

Therefore,

$$I(Z'; Y'_m) \geq I(Z'; Y'_s). \tag{24}$$

Combining this with the DPI result

$$I(Y'_s; Y'_m) \leq I(Y'_s; Z') = I(Z'; Y'_s),$$

we obtain the final inequality as

$$I(Z'; Y'_m) \geq I(Y'_s; Y'_m). \tag{25}$$

## A.2 Proof of prediction error bounds of the main task

### A.2.1 Binary case

For the two-class problem, the classifier predicts the input $z'$ with the posterior $\eta(z') = P\{Y'_m = 1 \mid Z' = z'\} > 0.5$ as positive, the minimum prediction error is

$$p(e) = \int_{Z'} \min\{\eta(z'), 1 - \eta(z')\} dp(z'). \tag{26}$$

Then Shannon entropy for a binary random variable with the distribution $(\eta, 1 - \eta)$ is defined as

$$H(\eta) = -\eta \cdot \log \eta - (1 - \eta) \cdot \log(1 - \eta), \quad \eta \in [0, 1]. \tag{27}$$

The expectation of the above function with respect to $z' \sim Z'$ is

$$H(Y'_m \mid Z') = \mathbb{E}_{z' \sim Z'}[H(\eta(z'))] = \int_{Z'} H(\eta(z')) dp(z'). \tag{28}$$

Based on Fano's inequality, we have $H_{\text{err}}(p(e)) \geq H(Y'_m \mid Z')$. As $p(e) \leq 0.5$ and the function $H_{\text{err}}(p(e))$ is monotonically increasing for $0 \leq \eta \leq 0.5$, we have

$$p(e) \geq H_{\text{err}}^{-1}(H(Y'_m \mid Z')). \tag{29}$$

Given that $H(Y'_m \mid Z') = H(Y'_m) - I(Z'; Y'_m)$ and $H(Y'_m \mid Y'_s) = H(Y'_m) - I(Y'_s; Y'_m)$, in the ideal test-time training under our Markov chain assumption $Y'_s \rightarrow Z' \rightarrow Y'_m$, $Z'$ is as informative about $Y'_m$ as $Y'_s$ is, we have

$$H(Y'_m \mid Z') = H(Y'_m \mid Y'_s), \tag{30}$$

then the lower bound of $p(e)$ is obtained as

$$p(e) \geq H_{\text{err}}^{-1}(H(Y'_m \mid Y'_s)). \tag{31}$$

Under Hellman's inequality (Hellman & Raviv, 1970), we have

$$p(e) \leq \frac{1}{2} H(Y'_m \mid Z'), \tag{32}$$

given $I(Z'; Y'_m) \geq I(Y'_s; Y'_m)$, the upper bound of $p(e)$ is derived as

$$p(e) \leq \frac{1}{2} H(Y'_m \mid Y'_s). \tag{33}$$

### A.2.2 MULTI-CLASS CASE

In a multi-class scenario, we assume $k$ classes, denoted as $\{1, \ldots, k\}$, and the main head classifier $\hat{Y}'_m$ maps the input $z' \in Z'$ to one of these $k$ classes. For $\eta = [\eta_1, \ldots, \eta_k]$, we have

$$h(\eta) = - \sum_{Y'_m=1}^{k} \eta_{Y'_m} \log \eta_{Y'_m}, \tag{34}$$

$$H(Y'_m \mid Z') = \mathbb{E}_{z' \sim Z'}[h(\eta(z'))] = \int_{Z'} h(\eta(z'))dp(z'), \tag{35}$$

$$p(e) = P(Y'_m \neq \hat{Y}'_m) = 1 - \sum_{Y'_m=1}^{k} \int_{Z'} \mathbb{1}_{\{Y'_m = \hat{Y}'_m\}} \eta_{Y'_m}(z')dp(z') = 1 - \mathbb{E}_{z' \sim Z'}\left[\max\{\eta(z')\}\right]. \tag{36}$$

Based on the simplified Fano's inequality (Thomas & Joy, 2006), we have

$$\begin{aligned} p(e) &\geq \frac{H(Y'_m \mid Z') - 1}{\log(|Y'_m|)} \\ &\geq \frac{H(Y'_m \mid Z') - 1}{\log(k)} \end{aligned} \tag{37}$$

Similar to the derivation in binary base, we can obtain

$$p(e) \geq \frac{H(Y'_m \mid Y'_s) - 1}{\log(k)}, \tag{38}$$

and when $k \geq 4$, the following always holds:

$$\frac{H(Y'_m \mid Y'_s) - 1}{\log(k)} \leq p(e) \leq \frac{1}{2} H(Y'_m \mid Y'_s). \tag{39}$$

## B DATASET, MODEL AND BASELINES

### B.1 DATASET

**Patient inclusion and exclusion criteria**. Patients were included in the respiratory failure prediction analysis if they had an ICU stay of at least five hours, were not mechanically ventilated before ICU admission, and had documented vital signs and laboratory values prior to the prediction start time. Those with a Do Not Resuscitate (DNR) order were excluded, and data within 24 hours of surgery were omitted to avoid bias from surgery-related ventilation. Monitoring continued until mechanical ventilation was initiated or ICU discharge. To ensure sufficient data, predictions began four hours post-admission and were updated hourly using the latest clinical information.

**Data abstraction and processing**. We extracted EHR data encompassing 50 vital signs and laboratory measurements, 6 demographic features, 12 Systemic Inflammatory Response Syndrome (SIRS) and Sequential Organ Failure Assessment (SOFA) criteria, 12 medication categories, and 62 comorbidities. To handle varying sampling frequencies, vital signs and laboratory values were aggregated into hourly time-series bins, with multiple measurements per hour summarized using the median. Data updates occurred hourly, with missing values carried forward for up to 24 hours if no new data were available. Remaining missing values were imputed using the mean. Additionally, we derived 150 features from the 50 vital signs and laboratory measurements, including baseline values (mean over the previous 72 hours), local trends (change since the last measurement), and time since last measured (TSLM).

**Clinical labeling scheme for the various physiological states of respiratory failure**. Table 3 lists the labeling criteria used in (Lam et al., 2024).

**Encounter-level evaluation**. Table 4 lists the details of evaluation metrics.

Table 3: Criteria of clinical labeling scheme.

| Condition | Criteria | Points |
|---|---|---|
| PaO$_2$/FiO$_2$ (not NaN) | $200 < \text{PaO}_2/\text{FiO}_2 \leq 300\,\text{mmHg}$ | 1 |
| | PaO$_2$/FiO$_2$ $\leq 200\,\text{mmHg}$ (severe hypoxemia) | 2 |
| | IMV $\leq 24\,\text{hours}$ | 3 |
| | PaO$_2$/FiO$_2$ $\leq 200\,\text{mmHg}$ and IMV $\leq 24\,\text{hours}$ | 4 |
| | IMV $> 24\,\text{hours}$ | 5 |
| SpO$_2$/FiO$_2$ (not NaN) | $141 < \text{SpO}_2/\text{FiO}_2 \leq 221\,\text{mmHg}$ | 1 |
| | SpO$_2$/FiO$_2$ $\leq 141\,\text{mmHg}$ (severe hypoxemia) | 2 |
| | IMV $\leq 24\,\text{hours}$ | 3 |
| | SpO$_2$/FiO$_2$ $\leq 141\,\text{mmHg}$ and IMV $\leq 24\,\text{hours}$ | 4 |
| | IMV $> 24\,\text{hours}$ | 5 |

Table 4: Definitions of evaluation metrics based on predictions and labels.

| Metric | Definition |
|---|---|
| **True Positive (TP)** | A positive prediction (predictions[t] $\geq$ threshold) where there is at least one positive label within the prediction window (up to 24 hours before $T_0$[a]). |
| **False Positive (FP)** | A positive prediction (predictions[t] $\geq$ threshold) where no positive labels exist within the prediction window (up to 24 hours before $T_0$). |
| **False Negative (FN)** | A negative prediction (predictions[t] $<$ threshold) where a positive label exists within the prediction window (up to 24 hours before $T_0$). |
| **True Negative (TN)** | A negative prediction (predictions[t] $<$ threshold) where no positive labels exist throughout the evaluated timestamps. |

[a] $T_0$ is defined as the first timestamp where a patient is ventilated based on the simultaneous recording of PEEP and FiO$_2$.

## B.2 NETWORK ARCHITECTURE

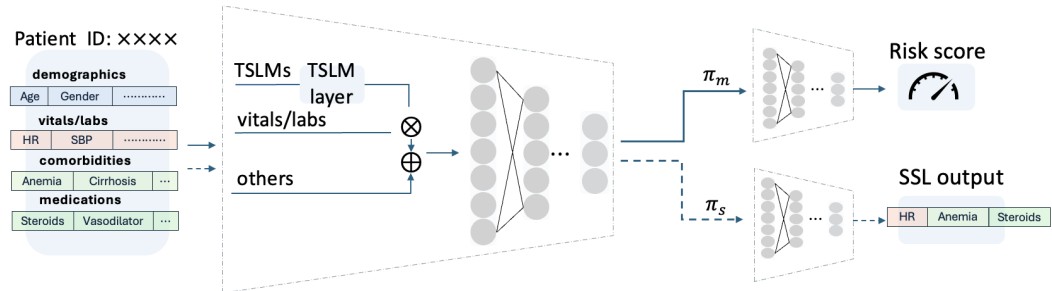

Figure 6: The developed network architecture. The encoder consists of a TSLM Layer followed by a feedforward neural network. Both main head and SSL head are feedforward neural networks

Our model follows a Y-shaped design with a shared encoder and two task heads (Fig. 6). At each hourly timestamp we assemble a structured input comprising static demographics, comorbidities/medications, and time-varying vitals/labs augmented with their time-since-last-measurement (TSLM). A lightweight *TSLM layer* ingests each raw value $x_j$ and its recency $\Delta t_j$, applying a learnable decay/gating function to down-weight stale observations and inject a recency embedding; the resulting features are concatenated with the static covariates and passed to a multilayer perceptron encoder $f_e(\cdot; \theta_e)$ to produce a latent representation $z \in \mathbb{R}^d$. Two shallow feedforward heads operate on $z$: a main classifier $\pi_m(\cdot; \theta_c)$ outputs the probability of IMV within 24 h, and a self-supervised head $\pi_s(\cdot; \theta_s)$ supports reconstruction and masked-feature modeling driven by our

dynamic, feature-aware masking scheme. At deployment the classifier and SSL head are frozen, and for each test example we adapt only the encoder for a few gradient steps; the adapted encoder is then used to produce the final risk score, after which weights are reset before the next instance.

## B.3 BASELINES

We compare against representative TTA/TTT methods and adapt each fairly to the EHR setting: **TEST**, **TENT** (Wang et al., 2020), **TTT** (Sun et al., 2020), **TTT++** Liu et al. (2021), **ClusT3** (Hakim et al., 2023), **NC-TTT** (Osowiechi et al., 2024), **T3A** (Iwasawa & Matsuo, 2021), **SAR** (Niu et al., 2023), and **CoTTA** (Wang et al., 2022).

**Fairness and EHR-specific protocol.** All methods use the same data pipeline (hourly aggregation, carry-forward $\leq$24h, mean imputation, and TSLM features), the same shared encoder as ours (Sec. B.2), and identical source-domain pretraining (optimizer, weight decay, early stopping). At deployment we enforce the source-free constraint (no source samples/labels). For gradient-based methods we fix the same test-time budget: five update steps, identical step size schedule, and reset-to-pretrained after each instance; the classifier head is frozen unless the baseline explicitly modifies it. Methods that require batch statistics (e.g., BN-based TTA) use the same first-in–first-out buffer of recent test samples to compute moments; the buffer size and confidence thresholds are tuned on the development validation split under the same hyperparameter budget (Bayesian optimization) for all methods. For approaches that rely on data augmentation (e.g., CoTTA), we replace image transforms with tabular augmentation: feature masking.

**Methods.**

- **TEST**: evaluate the pretrained model with no adaptation.

- **TENT** (Wang et al., 2020): minimize prediction entropy at test time; we update only BN affine parameters and running statistics.

- **TTT** (Sun et al., 2020): jointly train the network on the main task and the *same* auxiliary SSL objective as ours (reconstruction + masked–feature modeling) in the source domain; at test time, adapt the *encoder only* by minimizing this SSL loss. For fairness, TTT uses *uniform*, feature-agnostic masking (no dynamic masking), and does not use prototypes or optimal transport.

- **TTT++** (Liu et al., 2021): identical SSL setup as TTT above and the same encoder-only test-time updates; additionally aligns first/second-order moments between source and target. Because source data are unavailable at deployment, source moments are cached from the development split during training and used for alignment at test time.

- **ClusT3** (Hakim et al., 2023): add a projector on top of the shared encoder and adapt by maximizing mutual information with discrete codes. We use an MLP projector (tabular analogue of the original CNN projector) and the same codebook size across sites.

- **NC-TTT** (Osowiechi et al., 2024): optimize a noise-contrastive auxiliary likelihood at test time. The noise distribution is factorized Gaussian with per-feature mean/variance estimated on the development split.

- **T3A** (Iwasawa & Matsuo, 2021): optimization-free classifier adjustment that builds class prototypes from confident test predictions and reweights logits. We compute prototypes from the FIFO buffer, with the confidence threshold tuned on the development split; no backprop or encoder change.

- **SAR** (Niu et al., 2023): sharpness-aware entropy minimization with unreliable-sample filtering for small/test-time batches. We follow the BN-only update rule as in TENT, add SAM-style perturbations to BN parameters, and use the same FIFO buffer; filter thresholds are validated once on the development split.

- **CoTTA** (Wang et al., 2022): maintain an EMA teacher and perform augmentation/weight-averaged pseudo-labeling with periodic weight restoration. Tabular augmentation is feature masking; teacher momentum and restoration period are tuned under the shared budget. Encoder is adapted.

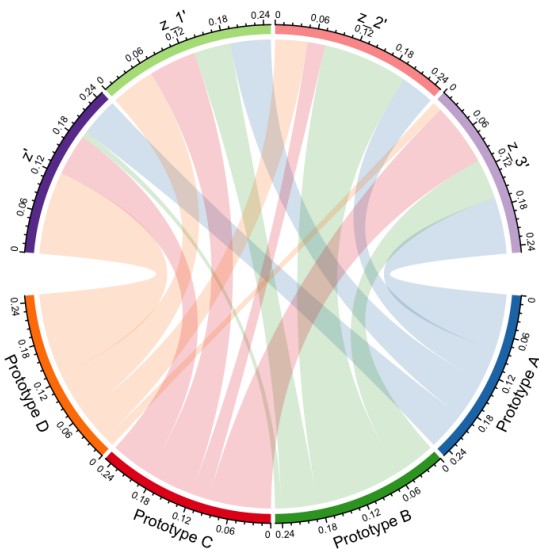

Figure 7: An example of POT between prototypes $\mathbf{P}$ and $z'$.

# C   ADDITIONAL RESULTS AND FIGURES

## C.1   AN EXAMPLE OF PARTIAL OPTIMAL TRANSPORT (POT)

To further illustrate how our method aligns test-time features with learned prototypes, we visualize an example of Partial Optimal Transport (POT) in Figure 7. The chord diagram shows the transport plan $\gamma$ between the test-time features $z'$ (top half of the circle) and prototypes $\mathbf{P}$ (bottom half). The width of each arc reflects the transported mass $\gamma_{ij}$ between feature $z'_i$ and prototype $p_j$.

Unlike fixed alignment methods, our formulation allows flexible, soft alignment by augmenting the test-time features with perturbed copies to enable better adaptation to test-time distribution shifts, which prevents overfitting to noisy test samples while preserving meaningful prototype relationships.

## C.2   ADDITIONAL SENSITIVITY ANALYSIS.

**Comparison of the size of prototypes**. Figure 8 shows the AUC performance for UCSDH, UCIH and MIMIC-IV across different prototype sizes. UCIH and MIMIC-IV exhibit a slight increase in performance as the prototype size increases, while UCSDH maintains relatively stable AUC values with minimal variation. UCIH and MIMIC-IV have more diverse underlying data distribution and while UCSDH may have more homogeneous patterns. Given that the complexity of external cohorts is unknown in advance, model calibration may be necessary to ensure optimal generalization.

**Comparison of Prototype Learning.** Our framework leverages prototypes to capture the training domain distribution and facilitate more effective alignment with test-time representations. To assess the contribution of prototype learning, we conduct two ablation studies.

First, as reported in Table 1 of the manuscript, we evaluate the PriTTT baseline, which removes the adaptive distribution matching component, thereby isolating the effect of prototype-guided alignment. Second, we examine the impact of prototype learning by replacing end-to-end learned prototypes with post hoc cluster centroids. Specifically, we extract training representations after model training and apply $k$-means clustering. Each cluster is represented by the average embedding of its members, and the resulting $k$ centroids are used as fixed prototypes for distribution matching during test-time training.

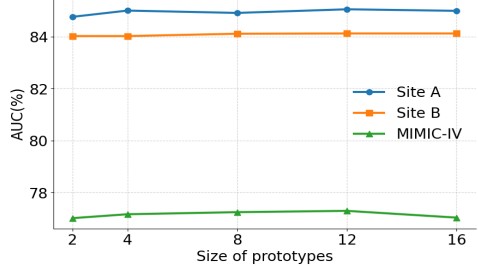

Figure 8: Effect of prototype size on AUC performance across different sites.

The results in Table 5 show that models using learned prototypes consistently outperform their post hoc counterparts across all datasets, which highlights the benefit of joint prototype learning and alignment in capturing richer, task-relevant training distributions.

Table 5: Performance Comparison with and without Prototype Learning.

| Method | UCSDH | UCIH | MIMIC |
|---|---|---|---|
| AdaTTT | $85.02 \pm 0.05$ | $84.10 \pm 0.05$ | $77.17 \pm 0.08$ |
| DynTTT | $84.54 \pm 0.10$ | $83.84 \pm 0.12$ | $76.79 \pm 0.05$ |
| AdaTTT w/o proto-learn | $84.57 \pm 0.04$ | $84.05 \pm 0.06$ | $76.85 \pm 0.07$ |
| DynTTT w/o proto-learn | $84.35 \pm 0.07$ | $83.69 \pm 0.08$ | $77.05 \pm 0.10$ |

Table 6: Ablation on masking strategy and SSL objectives (AUC %, ↑ higher is better).

| Dataset | TEST | Random Masking | Reconstruction Only | Dynamic Mask Only | AdaTTT (Full) |
|---|---|---|---|---|---|
| UCSDH | 84.01 | $82.45 \pm 0.05$ | $83.53 \pm 0.02$ | $84.71 \pm 0.02$ | $\mathbf{85.02 \pm 0.05}$ |
| UCIH | 83.75 | $82.60 \pm 0.06$ | $82.47 \pm 0.05$ | $83.18 \pm 0.04$ | $\mathbf{84.10 \pm 0.05}$ |
| MIMIC-IV | 75.28 | $74.21 \pm 0.04$ | $75.00 \pm 0.04$ | $76.24 \pm 0.02$ | $\mathbf{77.17 \pm 0.08}$ |

Table 7: Unified hyperparameter sensitivity analysis. All hyperparameters produce small variations (0.1–0.3 AUC), confirming robustness of AdaTTT.

| Hyperparameter Setting | UCSDH | UCIH | MIMIC |
|---|---|---|---|
| $\lambda_{\text{recon}} = 0.1$ | $85.01 \pm 0.05$ | $84.12 \pm 0.06$ | $77.16 \pm 0.08$ |
| $\lambda_{\text{recon}} = 0.5$ | $85.03 \pm 0.05$ | $84.11 \pm 0.05$ | $77.18 \pm 0.08$ |
| $\lambda_{\text{recon}} = 1.0$ | $85.00 \pm 0.06$ | $84.09 \pm 0.06$ | $77.14 \pm 0.09$ |
| $\lambda_{\text{proto}} = 0.1$ | $85.00 \pm 0.05$ | $84.08 \pm 0.06$ | $77.15 \pm 0.08$ |
| $\lambda_{\text{proto}} = 0.5$ | $85.02 \pm 0.05$ | $84.10 \pm 0.05$ | $77.17 \pm 0.08$ |
| $\lambda_{\text{proto}} = 1.0$ | $84.98 \pm 0.06$ | $84.07 \pm 0.06$ | $77.12 \pm 0.08$ |
| $\lambda_{\text{reg}} = 0.1$ | $85.01 \pm 0.05$ | $84.11 \pm 0.05$ | $77.16 \pm 0.08$ |
| $\lambda_{\text{reg}} = 0.5$ | $85.00 \pm 0.05$ | $84.09 \pm 0.05$ | $77.15 \pm 0.08$ |
| $\lambda_{\text{reg}} = 1.0$ | $84.97 \pm 0.06$ | $84.05 \pm 0.06$ | $77.13 \pm 0.09$ |
| $\lambda_{\text{ot}} = 0.1$ | $85.02 \pm 0.05$ | $84.11 \pm 0.06$ | $77.18 \pm 0.08$ |
| $\lambda_{\text{ot}} = 0.5$ | $85.04 \pm 0.05$ | $84.12 \pm 0.05$ | $77.20 \pm 0.08$ |
| $\lambda_{\text{ot}} = 1.0$ | $85.01 \pm 0.06$ | $84.09 \pm 0.06$ | $77.16 \pm 0.09$ |
| Warm-up = 0 epochs | $84.98 \pm 0.06$ | $84.07 \pm 0.06$ | $77.14 \pm 0.09$ |
| Warm-up = 5 epochs | $85.02 \pm 0.05$ | $84.10 \pm 0.05$ | $77.17 \pm 0.08$ |
| Warm-up = 10 epochs | $85.03 \pm 0.05$ | $84.11 \pm 0.05$ | $77.18 \pm 0.08$ |
| Warm-up = 20 epochs | $85.00 \pm 0.05$ | $84.09 \pm 0.06$ | $77.15 \pm 0.08$ |

**Dynamic masking and SSL objectives.**

We ablate the masking strategy and the auxiliary objectives. Replacing dynamic, task-aware masking with random masking degrades AUC by $\sim$1.5–3.0 points across sites (e.g., UCSDH: $85.02\pm0.05 \rightarrow 82.45\pm0.05$; MIMIC-IV: $77.17\pm0.08 \rightarrow 74.21\pm0.04$). Using a single objective alone (*reconstruction-only* or *masked-feature-only*) underperforms the full setup, indicating complementary roles: reconstruction regularizes representations, while masked feature modeling encourages uncertainty-aware recovery of informative variables (Table 6).

**Unified hyperparameter sensitivity analysis.**

As shown in Table 7, varying $\lambda_{\text{recon}}$, $\lambda_{\text{proto}}$, $\lambda_{\text{reg}}$, $\lambda_{\text{ot}}$, and warm-up epochs produces only small fluctuations ($\leq$ 0.1–0.3 AUC) across all cohorts, confirming that AdaTTT is stable to these hyperparameter choices.

