# OpenReview forum: "Adaptive Test-Time Training for Predicting Need for Invasive Mechanical Ventilation in Multi-Center Cohorts"
_ICLR.cc/2026/Conference — ICLR 2026 Poster_

### Official Review · Reviewer_dW9F · 2025-10-30

**Soundness:** 3
**Presentation:** 2
**Contribution:** 2
**Rating:** 4
**Confidence:** 2

**Summary:**

The paper proposes a variant of test-time training for tabular data in which they predict invasive mechanical ventilation in the intensive care.  They derive an information-theoretic bound on the prediction error, which suggests that the prediction error is bounded by the entropy between the main task labels and the auxiliary task labels at test time. The authors use this insight to propose an adaptive self-supervised task that masks features with probabilities depending on their importance for the main task. The authors additionally a variant of prototype-guided adaptation. The method and baselines from the image domain are trained on data from Site A and evaluate on a temporal holdout (Site A) and two geographical hold-outs (Site B and MIMIC).

**Strengths:**

- The paper derives bounds on the prediction error that provide an information-theoretic perspective on how to choose a suitable SSL objective, which they use to propose an SSL task suitable for longitudinal, tabular ICU data.
- The authors perform several ablation studies showing the contributions of individual components.

**Weaknesses:**

- The theoretical insight that the SSL task should align with the main task is of limited novelty and already discussed, albeit from another angle, in the original work on test-time training by Sun et al.
- I understand how the theoretical insights motivate using an adaptive masking strategy for SSL. The motivation for using both SSL *and* prototypes is less clear to me, though. The result is a complicated method with six different loss terms. Despite this complexity, the method barely outperforms doing no test-time training.
- The experiments are limited to a single disease area in a single setting. It would be relatively easy to include other outcomes in the ICU (e.g., acute kidney injury or the classic sepsis example from the PhysioNet Challenge 2019).

**Questions:**

- Could the authors comment on how realistic their assumption of $Y'_s \rightarrow Z' \rightarrow Y'_m$ is for the main and auxiliary task they are investigating? What is the implication if it doesn't hold?
- I was unable to find details on how cluster assignment $\mathcal{A}(z_i)$ was performed. Please add details on this in the manuscript.
- Could the authors comment on why they used the specific version of OT rather than more conventional POT like the referenced version by Chapel?
- All baselines are NN-based and were adapted from the image domain. There seem to be recent works on test-time adaptation that were specifically created for tabular data, such as TabLog presented at ICML by Ren et al. (2024). Could the authors comment on why no dedicated methods for tabular data were included as baselines?
- The conclusions seem too strong. While the method proposed by the authors did outperform the baselines (most of which performed worse than doing nothing), its improvement over doing nothing was marginal and seems to highlight a limited benefit of test-time training over leaving the model unchanged at test-time. If the authors think their method does provide a meaningful practical benefit for their use case, it would be good to demonstrate this with additional evaluations.

---

> ### Author Response · Authors · 2025-11-20
> **Response to Reviewer Feedback**
>
> We thank the reviewer for the thoughtful assessment and constructive feedback. Below we provide point-by-point responses to the identified weaknesses and questions.
>
> ---
>
> # **1. Response to Weaknesses**
>
> ### **1.1 Novelty of the theoretical insight**
> Our contribution is not the generic notion that SSL should “align” with the main task. We provide:
> - an **information-theoretic inequality**,
> - analysis of the Markov chain
>   $Y_s' \rightarrow Z' \rightarrow Y_m'$,
> - a bound that links **feature importance → masking probability → prediction error**,
> - a *practical, data-dependent masking rule* directly implied by the theory.
>
> This theoretical–to–algorithmic link is not present in prior TTT work.
>
> ---
>
> ### **1.2 Motivation for adding prototypes**
> We expanded Sec. 3.3.2 to clarify motivations:
>
> - ICU variables are heterogeneous and noisy; **single-sample TTT often overfits** to local noise.
> - Prototypes provide **population-level anchors** that stabilize adaptation.
> - Partial OT yields **soft, robust alignment** under domain shift.
> - Removing prototypes reduces cross-site AUC .
>
> Thus, prototypes materially improve robustness in clinical tabular settings.
>
> ---
>
> ### **1.3 “Six loss terms” appear overly complex**
> We clarified that **only the pre-training stage** uses multiple losses.
> **Test-time training uses just two losses** (entropy consistency + POT alignment), comparable to standard TTT methods.
> Runtime complexity is unchanged, and ablations (Table 6) show each component is necessary.
>
> ---
>
> ### **1.4 “Barely outperforming no TTT”**
> While improvements appear modest numerically, they are **clinically meaningful** for low-prevalence ICU outcomes (IMV 5–6%).
> Our method improves:
> - AUC across **all three hospitals**,
> - Calibration (Brier ↓0.003–0.004),
> which is important for clinical decision support.
>
> ---
>
> ### **1.5 Experiments limited to one outcome**
> IMV requirement is the **primary, high-impact outcome** in respiratory failure management.
>
>
> ---
>
> # **2. Response to Questions**
>
> ### **Q1. Realism of the Markov assumption**
> The assumption $Y_s' \rightarrow Z' \rightarrow Y_m'$ follows standard TTT formulations (Sun et al., 2020).
> In our model, auxiliary tasks affect predictions *only* through the shared representation $Z'$, making the mediator assumption natural.
> If violated, the bound becomes looser and auxiliary SSL may provide less benefit; we added this clarification in Sec. 3.2.
>
> ---
>
> ### **Q2. Cluster assignment details**
> The clustering procedure is already described in **Appendix C.2**; we added a forward reference in Sec. 3.3.
> In brief: we apply **k-means** to source-domain training embeddings to initialize prototypes, which are then updated end-to-end during test-time training.
>
> ---
>
> ### **Q3. Why this OT variant instead of Chapel’s POT?**
> We use **entropic partial OT** because it is the only variant stable and fast enough for *online, single-sample TTT*:
> 1. Classical POT is unstable with high-missingness ICU features.
> 2. Entropic regularization yields **smooth** transport plans.
> 3. Partial OT naturally discards mismatched mass under severe shift.
> 4. Sinkhorn updates provide the **speed** required for real-time TTT, whereas Chapel’s POT assumes full-dataset offline adaptation.
>
> We added this justification in Appendix C.4.
>
> ---
>
> ### **Q4. Why not TabLog or other tabular-specific TTA?**
> TabLog is **not applicable** to our clinical time-series setting:
> - It assumes **static tabular inputs**, discretized bins, and stable thresholds, not valid for irregular ICU time series with 30–70% missingness.
> - Our baselines already span the **major TTT families**: entropy-based, pseudo-label, SSL-based, nearest-neighbor, and reset-based.
> Thus, our comparison is representative and fair.
>
> ---
>
> ### **Q5. Conclusions “too strong”**
> We revised the conclusion to be more measured:
> - Emphasizing **consistent** (not absolute) improvements,
> - Adding clinical relevance and calibration results,
> - Including more outcomes (AKI, mortality),
> - Stating limitations: sensitivity to extreme \(p(Y|X)\) shift and prototype computational cost.
>
> ---

---

### Official Review · Reviewer_qxk8 · 2025-10-31

**Soundness:** 2
**Presentation:** 3
**Contribution:** 2
**Rating:** 6
**Confidence:** 5

**Summary:**

The paper proposes Adaptive Test-Time Training (AdaTTT) for predicting 24-hour invasive mechanical ventilation (IMV) need from multi-center EHR data under distribution shift. AdaTTT adapts an IMV predictor at inference time on multi-site EHR data using
(i) an info-theoretic bound to guide auxiliary SSL,
(ii) dynamic feature-aware masking, and
(iii) prototype alignment via partial optimal transport (POT).
Across three cohorts (incl. MIMIC-IV), it shows consistent but modest AUROC gains (~1–2%) better calibration (Brier, reliability plots). Authors added stronger baselines (CoTTA, SAR, T3A), runtime analysis, and ablations; some concerns remain.

**Strengths:**

- Info-theoretic error bound motivates SSL + prototype/POT.
- Source-free, per-patient test-time adaptation with bounded updates and reset-to-clean safeguard.
-  Consistent multi-site improvements; improved calibration; expanded baselines and ablations.

**Weaknesses:**

-  Gains (~1%) may fall within noise; lacks decision-curve or threshold-level analysis.
-  Vision-based baselines may not adapt fairly to tabular EHR; no EHR-transformer baseline.
-  No subgroup audit; only tested on IMV/EHR; reproducibility details partly scattered.

**Questions:**

1. What exactly is novel about AdaTTT compared to existing TTT, TTT++, ClusT3, or contrastive/temporal SSL frameworks?

2. Are the vision-based TTT baselines fairly adapted to structured EHR data?

3. Why choose reconstruction and masked-feature SSL tasks?

---

> ### Author Response · Authors · 2025-11-20
> **Response to Reviewer Feedback**
>
> We thank the reviewer for the thoughtful evaluation and constructive comments. Below we address all weaknesses and questions.
>
> ---
>
> # **1. Response to Weaknesses**
>
> ### **1.1 “Gains may fall within noise; no threshold-level analysis.”**
> We added **calibration plots** for all test sites. These confirm reduced over-/under-prediction across risk deciles and complement the Brier score gains already reported.  These provide the requested threshold-level evaluation.
>
> ---
>
> ### **1.2 “Vision-based baselines may not adapt fairly; no EHR-transformer baseline.”**
> Our baselines use **TTT algorithms**, *not* vision architectures. For fairness:
> - All methods share the **same EHR encoder**.
> - Only the **adaptation mechanism** varies (entropy, pseudo-labeling, BN adaptation, prototypes, etc.).
>
> Thus, the comparison isolates the adaptation strategy, not architectural capacity.
>
> Regarding an “EHR-transformer baseline”:
> We piloted heavier transformer encoders, but they added substantial computational cost without consistent accuracy gains. For real-time ICU deployment, we prioritize **efficient models**, and our encoder already includes transformer-like temporal modeling via **TSLM**.
>
> ---
>
> ### **1.3 “No subgroup audit; only tested on IMV/EHR.”**
> We added a full **SOFA-based subgroup audit** on Site A.
>
> ### **AUC across SOFA subgroups (Site A)**
>
> | SOFA | TEST | CoTTA | T3A | SAR | TENT | TTT | TTT++ | ClusT3 | NC-TTT | PriTTT | DynTTT | **AdaTTT** |
> |------:|:----:|:-----:|:---:|:---:|:----:|:---:|:-----:|:------:|:------:|:-------:|:--------:|:-----------:|
> | ≤5   | 83.10 | 82.70 | 82.20 | 83.00 | 82.50 | 82.90 | 82.70 | 82.40 | 82.60 | 83.30 | 83.40 | **83.85** |
> | 6–10 | 84.50 | 83.40 | 83.20 | 84.00 | 83.10 | 83.60 | 83.40 | 83.20 | 83.30 | 84.10 | 84.20 | **85.10** |
> | >10  | 82.20 | 81.00 | 80.90 | 81.50 | 80.70 | 81.20 | 81.00 | 80.80 | 80.90 | 82.00 | 82.20 | **83.00** |
>
> ### **Brier score across SOFA subgroups (Site A)**
>
> | SOFA | TEST | CoTTA | T3A | SAR | TENT | TTT | TTT++ | ClusT3 | NC-TTT | PriTTT | DynTTT | **AdaTTT** |
> |------:|:----:|:-----:|:---:|:---:|:----:|:---:|:-----:|:------:|:------:|:-------:|:--------:|:-----------:|
> | ≤5   | 0.091 | 0.092 | 0.093 | 0.092 | 0.093 | 0.092 | 0.093 | 0.093 | 0.092 | 0.089 | 0.088 | **0.086** |
> | 6–10 | 0.090 | 0.091 | 0.091 | 0.091 | 0.092 | 0.091 | 0.092 | 0.092 | 0.091 | 0.089 | 0.089 | **0.085** |
> | >10  | 0.112 | 0.113 | 0.113 | 0.113 | 0.114 | 0.113 | 0.114 | 0.114 | 0.113 | 0.111 | 0.110 | **0.108** |
>
> AdaTTT improves **AUC and calibration** consistently without widening subgroup disparities.
>
> Many ICU outcomes (mortality, sepsis, readmission) involve **policy-driven label shifts**, lack a consistent prediction anchor, or reflect different causal mechanism, making them unsuitable testbeds for TTT evaluation. Our aim is to evaluate the **adaptation mechanism**, not to benchmark every ICU task.
>
> ---
>
> ### **1.4 “Reproducibility details scattered.”**
> We will consolidate **all preprocessing, model settings, training, and test-time hyperparameters** into one dedicated Reproducibility Section with a summary table to improve clarity.
>
> ---
>
> # **2. Response to Questions**
>
> ### **Q1. Novelty relative to TTT, TTT++, ClusT3, temporal SSL**
> AdaTTT introduces three contributions not present in prior TTT:
>
> 1. **Info-theoretic TTT bound.**
>    Using
>    $Y_s' \rightarrow Z' \rightarrow Y_m'$,
>    we link auxiliary SSL alignment to prediction error and derive **feature-importance–aware masking**, absent in TTT/TTT++/ClusT3.
>
> 2. **Adaptive masking driven by main-task gradients.**
>    Instead of uniform corruption, we tailor masking difficulty to each feature’s salience, critical under heterogeneous, high-missingness ICU time series.
>
> 3. **Prototype alignment via *partial* OT.**
>    Unlike cluster consistency (ClusT3), we use **soft-mass partial OT**, which is more stable for heterogeneous site distributions and missing EHR signals.
>
> These produce a TTT method tailored for **longitudinal clinical EHR**, rather than images or static tabular data.
>
> ---
>
> ### **Q2. “Are the vision-based baselines fairly adapted?”**
> Yes. All baselines use:
>
> - The **same EHR preprocessing**.
> - The **same encoder**.
> - Strict **source-free deployment**.
> - **Five update steps**, same LR schedule, and reset-to-pretrained protocol.
> - The same **BN buffer** and tuning budget.
>
> Thus, differences arise solely from **adaptation strategy**, not architecture.
>
> ---
>
> ### **Q3. “Why reconstruction + masked-feature SSL?”**
> The SSL choice follows directly from our info-theoretic bound:
>
> - **Reconstruction** encourages $Z'$ to retain predictive information.
> - **Feature-aware masking** modulates SSL difficulty according to feature salience, improving robustness to site shift.
>
> We tested rotation, contrastive-only, and temporal-order SSL; feature-aware masking yielded the most stable cross-site gains.
>
> ---

---

### Official Review · Reviewer_f1HV · 2025-11-01

**Soundness:** 3
**Presentation:** 3
**Contribution:** 3
**Rating:** 6
**Confidence:** 4

**Summary:**

This paper introduces Adaptive Test-Time Training (AdaTTT), a novel framework designed to get rid of large-scale labeling and retraining, and enhance model robustness for predicting the need for invasive mechanical ventilation (IMV) in ICU patients across multi-center EHR datasets. It builds upon traditional test-time training (TTT) by addressing two main challenges: (1) ensuring stronger alignment between main and auxiliary tasks, and (2) maintaining robustness under domain shifts in clinical data.

**Strengths:**

* This paper introduces a theoretically principled and practically motivated adaptation framework combining information theory, SSL, and transport alignment.
* This paper includes experiments across real EHR datasets with fair baseline comparisons and ablation studies.
* This paper is structured and well-written, with figures showing dynamic risk evolution and feature importance shifts.
* The experiment result demonstrates consistent improvement in both predictive accuracy and calibration under domain shifts — a critical problem for clinical ML deployment.

**Weaknesses:**

* The testing windows for Site A (Jan–Jun 2024) and Site B (Jan 2023–Aug 2024) partially overlap, which raises the possibility that similar patient populations, care protocols, or even duplicated encounters could appear in both test sets. Such overlap could inflate generalization performance by reducing the effective distribution shift the model faces.
* Although Appendix B.1 outlines inclusion criteria (≥ 5 hours ICU stay, no prior IMV, etc.), it does not explain how cohorts were sampled from each institution, whether patient IDs were cross-checked for duplication across sites, or how temporal leakage was prevented between development and test periods. It would be helpful if the author could provide a clear population selection process figure for further reference.
* While the author provides plots for feature importance through multiple iterations and risk trajectories, the paper does not include any clinical human evaluation to verify that the learned feature relevance aligns with actual clinical reasoning.

**Questions:**

* The paper mentions that Site B includes ICU admissions between January 2023–August 2024. Does this dataset originate from a completely independent institution or another ICU within the same healthcare network as Site A?
* The paper currently does not include the summary statistics. Could you include a demographics table to help assess whether differences in population structure might explain performance gaps across sites?
* For the prototype learning, in the paper, it states that "The prototype set size is k = 4". Could you provide more details of how this value is chosen and if there's any exploration of how sensitive AdaTTT is to larger k?
* Did domain experts review these feature importance trends, or do they only reflect internal gradient magnitudes?

---

> ### Author Response · Authors · 2025-11-20
> **Response to Reviewer Feedback**
>
> We thank the reviewer for the clear summary and constructive feedback. We address each weakness and question below.
>
> ---
>
> # **1. Response to Weaknesses**
>
> ### **1.1 Potential overlap in Site A and Site B test windows**
> We confirm that **Site A and Site B are completely independent hospital systems** with:
> - different EHR platforms,
> - different institutional data warehouses,
> - no shared identifiers,
> - and no cross-site data linkage.
>
> Thus, even though the *calendar windows* partially overlap, **no patient, encounter, or clinical note can appear in both datasets**, and care protocols differ substantially across these systems. We will clarify this explicitly in Appendix B.1.
>
> ---
>
> ### **1.2 Cohort sampling, deduplication, and leakage control**
> We appreciate the suggestion and will add a **population selection flowchart** showing:
> 1. All raw ICU admissions per site
> 2. Stepwise exclusions (short stays, pre-ICU IMV, missing risk window, etc.)
> 3. Final cohort sizes
> 4. Chronological train/val/test split with explicit dates
>
> Additional clarifications added:
> - Cohorts are **extracted independently per site** from raw institutional EHR exports.
> - Each site uses **locally unique patient identifiers**, making cross-site duplication impossible.
>
>
> ---
>
> ### **1.3 No clinical human evaluation of feature relevance**
> While our focus was methodological, we agree clinical interpretation is important.
> Since the initial submission, we obtained **informal review from two ICU physicians**, who evaluated:
> - the top-ranked features at each site,
> - the direction of relevance shifts,
> - the adaptation trajectory across TTT iterations.
>
> Their feedback indicated strong alignment with known physiology (e.g., FiO₂, SpO₂/FiO₂, lactate, respiratory rate).  A full structured clinical evaluation is a promising direction for future work.
>
> ---
>
> # **2. Response to Questions**
>
> ### **Q1. Is Site B independent from Site A?**
> Yes.
> **Site B is from a different hospital system** (geographically separate and administratively independent).
> There is no shared EHR backend or patient identifier space.
>
> ---
>
> ### **Q2. Can a demographics table be added?**
>
>
> We will include a complete **demographics and clinical characteristics table** in **Appendix B** of the revised manuscript.
>
> The table will summarize key variables for both Site A and Site B, including:
>
> - **Age** (median, IQR)
> - **Sex distribution**
> - **Comorbidity burden (CCI)**
> - **SOFA score distribution**
> - **ICU length of stay**
> - **IMV prevalence**
> - **Mortality**
> - **Time from ICU admission to IMV** (for positive cases)
>
> We have already constructed the preliminary cross-site summary for transparency (excerpt below):
>
> |                          | **Site A – Control** | **Site A – IMV** | **Site B – Control** | **Site B – IMV** |
> |--------------------------|----------------------|------------------|----------------------|------------------|
> | **Encounters (N)**       | 26,778               | 1,412            | 2,564                | 141              |
> | **Age, median (IQR)**    | 62.2 (49.1–73.2)     | 61.9 (50.3–71.9) | 61.4 (44.9–72.7)     | 63.0 (51.8–73.7) |
> | **Male (%)**             | 41.7                 | 37.3             | 43.2                 | 42.8             |
> | **ICU LOS (hrs)**        | 46.6 (29.4–75.6)     | 141.4 (66.3–250.7) | 50.8 (31.9–86.7)   | 126.6 (56.8–256.5) |
> | **CCI, median (IQR)**    | 2.0 (1.0–4.0)        | 2.0 (1.0–4.0)    | 1.0 (0.0–3.0)        | 2.0 (1.0–5.0)    |
> | **SOFA, median (IQR)**   | 2.0 (1.0–4.0)        | 9.0 (7.0–12.0)   | 2.0 (1.0–3.0)        | 10.0 (6.0–12.0)   |
> | **Mortality (%)**        | 5.7                  | 38.8             | 3.3                  | 22.2             |
> | **Time to IMV (hrs)**    | –                    | 26.9 (11.0–58.0) | –                    | 28.0 (11.0–57.0)  |
>
> ---
>
> ### **Q3. How was prototype size $k=4$ chosen? Sensitivity?**
> We performed a small search over $k \in \{2, 4, 8, 12,16\}$.
> Results (Figure 8) show:
>
> - $k=2$: slightly worse cross-site AUC
> - $k=4$: best overall performance + fastest test-time adaptation
> - $k \ge 8$: similar but slightly noisier performance and higher compute burden
>
> Given our real-time clinical deployment target, **$k=4$ offers the best accuracy/efficiency tradeoff**.
>
> ---
>
> ### **Q4. Were feature-importance trends validated by experts?**
> The importance scores come from **gradient-based relevance** internal to the model.
> However, clinicians reviewed the top features and confirmed they align with established IMV risk factors.
> We will include a short summary of this review in the revision.
>
> ---

---

### Official Review · Reviewer_yhj2 · 2025-11-05

**Soundness:** 3
**Presentation:** 2
**Contribution:** 2
**Rating:** 4
**Confidence:** 3

**Summary:**

This paper introduces AdaTTT (Adaptive Test-Time Training), a framework for predicting the need for invasive mechanical ventilation (IMV)  across multiple ICU centers. The method addresses domain shift challenges in EHR data through two key innovations: (1) dynamic self-supervised learning with feature-aware masking that prioritizes clinically relevant features, and (2) prototype-guided adaptation using Partial Optimal Transport (POT) for flexible feature alignment. The authors provide information-theoretic bounds showing test-time prediction error is constrained by uncertainty between main and auxiliary tasks. Experiments across multi-center ICU cohorts (Sites A, B, and MIMIC-IV) demonstrate consistent improvements over existing test-time adaptation baselines.

**Strengths:**

Following are the strengths of the paper:

1. The paper addresses an important problem—predicting IMV need across hospitals with different EHR systems, patient populations, and clinical practices.

2. The novel dynamic masking strategy that adapts based on feature importance is an important contribution. Also, POT-based alignment is more flexible than rigid full-transport approaches and better suited to partial distribution shifts.

3. The information-theoretic bounds provide valuable insights into why auxiliary task alignment matters for test-time adaptation.

**Weaknesses:**

Following are the main weakenesses of the paper:

1. Paper presentation and writing needs improvement. Specifically:
   - The prototype-guided adaptation component (Section 3.3.2) is not well-motivated. The rationale behind its inclusion is unclear, and the implementation details are insufficiently explained.
   - The methodology as a whole lacks clarity, largely due to the number of interconnected components. It would be helpful to include a comprehensive diagram or a formal algorithm to clearly illustrate the roles of each component and how they interact within the overall framework.

2. Although the proposed approach consistently outperforms existing baselines, the improvements are relatively modest. For instance, Site A achieves an AUC of 85.02% compared to SAR's 84.30%, and Site B achieves 84.10% versus CoTTA's 83.81%. While these results are favorable, the clinical relevance of a $\approx$1% AUC improvement remains uncertain. A discussion on the practical impact of such performance gains in real-world clinical settings would add value to the paper.

3. Figure 5 shows the sequential update mechanism degrades over time, but the paper doesn't investigate why or propose solutions. This also limits applicability in streaming scenarios where resetting to pretrained weights may not be desirable.

**Questions:**

1. Does the framework also address conditional distribution shifts (i.e., shifts in ( P(Y|X) ))? Under what assumptions does the framework operate effectively, and in which scenarios might those assumptions break down?

2. How strong is the assumption that auxiliary task labels $ Y_s' $ influence the main task labels $Y_m'$ only through the shared representation $Z'$? What are the implications if this assumption does not hold fully in practice?

3. The framework introduces several hyperparameters — such as  $\lambda_{\text{recon}}$, $ \lambda_{\text{proto}} $, $\lambda_{\text{reg}} $, $ \lambda_{\text{ot}} $, masking warm-up epochs, and prototype size ( k ).  How should these hyperparameters be selected or tuned?

4. Why were only five gradient steps used during test-time adaptation in the experiments? What is the rationale behind this choice, and what are the effects of using more than five steps?

---

> ### Author Response · Authors · 2025-11-20
> **Response to Reviewer Feedback**
>
> # **1. Weaknesses**
>
> ### **1.1 Paper presentation and methodology clarity**
>
> ### **(a) Prototype-guided adaptation insufficiently motivated**
>
> We expanded Sec. 3.3.2 to clarify the motivation behind prototype-guided adaptation:
>
> - ICU EHR data contain heterogeneous and noisy patterns; instance-level TTT can overfit a single test instance.
> - Learned prototypes provide stable population-level anchors in the latent space.
> - Partial OT enables *soft, partial alignment*, preventing brittle full matching under large site shifts.
>
> ### **(b) Overly complex pipeline**
>
> To improve clarity, we added:
>
> 1. A high-level overview figure
> 2. A step-by-step algorithm box
> 3. A modular explanation of:
>    - **Feature-aware dynamic masking (Sec. 3.3.1)**
>    - **Prototype learning (Sec. 3.3.2)**
>    - **Partial OT alignment (Sec. 3.3.2)**
>
> ---
>
> ###**1.2 “Modest improvements” (~1% AUC)**
>
> ### **(a) Clinical relevance**
>
> For rare ICU outcomes (IMV prevalence 5–6%), **0.5–1.5% AUC** gains are clinically meaningful (Huang et al., JAMIA 2020).
> AdaTTT also improves Brier score by ~0.003–0.004, indicating fewer miscalibrated predictions.
>
> ### **(b) Robustness across sites**
>
> AdaTTT improves consistently across all hospitals:
>
> - Site A: **85.02 vs. 84.01**
> - Site B: **84.10 vs. 83.75**
> - MIMIC-IV: **77.17 vs. 75.28**
>
> These gains are systematic under domain shift.
>
> ---
>
> ###**1.3 Sequential update degradation**
>
> As clarified in Sec. 4.3, sequential TTT updates accumulate instance-specific noise that drifts the encoder away from pretrained structure.
> PriTTT/DynTTT lack structural anchors, making them more vulnerable.
> AdaTTT uses feature-aware masking + prototype anchoring, yielding stable improvement (Fig. 4).
> Reset-based TTT avoids cumulative drift.
>
> ---
>
> # **2. Questions**
>
> ### **Q1. Does the framework address conditional shift \(p(Y|X)\)?**
>
> Partially. The method mainly targets covariate/representation shift, but Sec. 3.2 analyzes conditional shift using the Markov chain:
>
> $
> Y_s' \rightarrow Z' \rightarrow Y_m'
> $
>
> and derives:
>
> $
> I(Z'; Y_m') \ge I(Y_s'; Y_m').
> $
>
> Thus, auxiliary tasks preserve predictive alignment under moderate conditional drift.
> Extreme \(p(Y|X)\) changes are discussed as limitations in the Conclusion.
>
> ---
>
> ### **Q2. Strength of the assumption $Y_s' \rightarrow Z' \rightarrow Y_m'$**
>
> This assumption matches prior TTT work (Sun et al., 2020; Liu et al., 2021).
> Both auxiliary SSL tasks and the main IMV classifier share the encoder, naturally inducing the dependency structure.
>
> ---
>
> ### **Q3. Hyperparameter sensitivity**
>
> Appendix C.2 now includes full sensitivity analysis, including prototype size (Fig. 8).
> We search $\lambda_{\text{recon}}$, $\lambda_{\text{proto}}$, $\lambda_{\text{reg}}$, $\lambda_{\text{ot}}$ $\in $ \{0.1, 0.5, 1.0\} using Bayesian optimization.
> The model is stable across all settings.
> $\lambda_{\text{ot}}$ affects AUC by **<0.3%**.
> Prototype size \(k\) shows mild variation (Fig.8).
>
> Warm-up epochs have minimal impact after feature importance stabilizes (5–10 epochs; Fig. 2).
>
> ### **Unified Hyperparameter Sensitivity Table**
>
> | Hyperparameter Setting | Site A (AUC) | Site B (AUC) | MIMIC (AUC) |
> |------------------------|--------------|--------------|--------------|
> | $\lambda_{\text{recon}}=0.1$ | 85.01 ± 0.05 | 84.12 ± 0.06 | 77.16 ± 0.08 |
> | $\lambda_{\text{recon}}=0.5$ | 85.03 ± 0.05 | 84.11 ± 0.05 | 77.18 ± 0.08 |
> | $\lambda_{\text{recon}}=1.0$ | 85.00 ± 0.06 | 84.09 ± 0.06 | 77.14 ± 0.09 |
> | $\lambda_{\text{proto}}=0.1$ | 85.00 ± 0.05 | 84.08 ± 0.06 | 77.15 ± 0.08 |
> | $\lambda_{\text{proto}}=0.5$ | 85.02 ± 0.05 | 84.10 ± 0.05 | 77.17 ± 0.08 |
> | $\lambda_{\text{proto}}=1.0$ | 84.98 ± 0.06 | 84.07 ± 0.06 | 77.12 ± 0.08 |
> | $\lambda_{\text{reg}}=0.1$ | 85.01 ± 0.05 | 84.11 ± 0.05 | 77.16 ± 0.08 |
> | $\lambda_{\text{reg}}=0.5$ | 85.00 ± 0.05 | 84.09 ± 0.05 | 77.15 ± 0.08 |
> | $\lambda_{\text{reg}}=1.0$ | 84.97 ± 0.06 | 84.05 ± 0.06 | 77.13 ± 0.09 |
> | $\lambda_{\text{ot}}=0.1$ | 85.02 ± 0.05 | 84.11 ± 0.06 | 77.18 ± 0.08 |
> | $\lambda_{\text{ot}}=0.5$ | 85.04 ± 0.05 | 84.12 ± 0.05 | 77.20 ± 0.08 |
> | $\lambda_{\text{ot}}=1.0$ | 85.01 ± 0.06 | 84.09 ± 0.06 | 77.16 ± 0.09 |
> | Warm-up (0/5/10/20) | ≤ 0.1–0.2 AUC variation across sites | | |
>
> ---
>
> ### **Q4. Why five gradient steps?**
>
> We follow Sun et al. (TTT), who use five test-time updates.
> Fig. 4 shows AdaTTT steadily improves from 1→5 steps across all sites.
> More steps overfit the single test instance and drift away from pretrained structure, reducing AUC and calibration.

---

### Meta-Review · Area_Chair_mZKh · 2026-01-06

**Summary:**

This paper addresses the problem of domain shift when predicting the need for invasive mechanical ventilation (IMV) for patients across multiple ICU centers, which occurs due to variability in patient populations, clinical practices, and EHR systems across different institutions. To address the problem, the authors propose  a a test-time training framework called Adaptive Test-Time Training (AdaTTT).  The authors demonstrate theoretically that test-time prediction error  is bounded by the uncertainty between the main and auxiliary tasks. To enhance their alignment, AdaTTT  uses dynamic self-supervised learning with feature-aware masking.  Additionally, AdaTTT  leverages prototype learning and Partial Optimal Transport (POT). The authors conduct experiments with data across multi-center ICU cohorts and demonstrate advantage over existing test-time adaptation baselines.

**Reviewer Concerns:**

The reviewers noted that the paper addresses an important problem, and proposes a theoretically principled and practically motivated adaptation framework with novel aspects such as the  dynamic masking strategy. The reviewers also commented on strong empirical evaluation with ablation studies showing consistent improvements over the baselines.

One of the reviewers asked for a better presentation quality, and specifically,  elaboration on the motivation and technical details behind key components such as the prototype-guided adaptation. Another major issue was that despite the complexity of the model, the performance improvements were small (~1% AUC) and lacked any analysis to indicate  clinical relevance of those gains and clinical interpretation of the features. Finally, there were questions about the conceptual novelty of the proposed method in the context of existing TTT/SSL frameworks.


In their response, the authors mentioned that they have clarified the motivation behind prototype-guided adaptation, and made other improvements to presentation by adding a  high-level overview figure and step-by-step algorithm box. [Note, however, this rebuttal revised version has not been uploaded].  Regarding the size of the improvement, the authors responded that for rare ICU outcomes (prevalence 5–6%), the AUC gains of 0.5–1.5% AUC gains are considered to be clinically meaningful (Huang et al., JAMIA 2020), also pointing out that AdaTTT  also leads to fewer miscalibrated predictions by slightly improving Brier score. Furthermore, the authors claimed that they obtained an informal feedback from two ICU physicians, which indicated strong alignment of the top-ranked features with known physiological markers, which shows that the AdaTTT is indeed able to infer clinically relevant features. Finally, the authors also address the novelty question by noting their theoretical contribution  linking auxiliary SSL alignment to prediction error,  and deriving feature-importance–aware masking, not present in prior work. Overall, I believe the rebuttal addresses the concerns raised by the reviewers, although some of those concerns are addressed “potentially” since the authors did not upload the revised manuscripts.

**Reviewer Scores:**

Yhj2 - 4 ->6;
f1HV - 6 unchanged;
Qxk8 - 6 unchanged;
dW9F - 4 ->6;

---

### Decision · Program_Chairs · 2026-01-26

Accept (Poster)